# Quantitative Research on the Form of Traditional Villages Based on the Space Gene—A Case Study of Shibadong Village in Western Hunan, China

**Zhongyi Nie [1], Ni Li [1,\*], Wei Pan [1], Yusheng Yang [2], Wei Chen [3] and Chenlei Hong [4]**

[1]  School of Architecture and Art, Central South University, Changsha 410083, China; 201312025@csu.edu.cn (Z.N.); 128148@csu.edu.cn (W.P.)
[2]  College of Architecture and Urban Planning, Tongji University, Shanghai 200092, China; yshy21@126.com
[3]  Changsha Urban Planning Information Service Center, Changsha 410221, China; 203911015@csu.edu.cn
[4]  Wuhan Institute of Technology, Wuhan 430205, China; hcl960429@126.com
\*  Correspondence: lini_zn@csu.edu.cn

**Abstract:** Traditional villages are the place where national culture is nurtured and inherited. Due to the acceleration of urbanization, the protective exploitation of traditional villages is an urgency in many regions of the world. Under the perspective of the "Space Gene", we define the connotation of the traditional village space gene, which refers not only to a space combination model, but also the basic inheritance unit of the sustainable development of traditional villages. We further propose the Quantitative Inheritance System Model of Traditional Village Space Genes, which reveals the logic behind the formation of traditional village material forms and builds a quantitative index system for traditional village forms. We take Shibadong Village as an example to prove the model. The results demonstrate that although the four camps in Shibadong Village belong to the same ethnic group and the same village, there are still apparent differences in morphological features. Through the model, we can turn from the "built form" research from the perspective of material space to the "deep structure" research behind it, which can provide scientific guidance for the planning and designs of traditional villages to inherit the history and culture and to protect the diversity of world culture.

**Keywords:** quantitative research; traditional village; form; space gene; western Hunan

## 1. Introduction

In the national cultural heritage protection system, landscape protection has always been a crucial issue [1]. Traditional villages have tangible and intangible cultural heritage, which are of immeasurable value in history, culture, science, art, society, and the economy [2]. The rural landscape is a sustainable cultural landscape [3]. Cultural diversity constitutes a colorful world, one of the sources of world development, and a vital driving force for sustainable development [4,5]. However, in today's world, with the deepening of globalization, especially in developing countries, the urbanization rate is getting higher and higher, and many significant changes have taken place in rural areas and rural production lifestyles. In urbanization and modernization, traditional agriculture and rural culture are disappearing or being assimilated [6]. Traditional villages face many problems such as constructive destruction, ecological environment pollution, aging of traditional buildings, and functional degradation, etc. [7]. Nowadays, China has made numerous efforts to protect traditional villages. Since the first batch of Chinese traditional village directories was announced in 2012, five batches of traditional village directories have been declared, involving 6819 traditional villages. Among them, Hongcun and Xidi are World Cultural Heritages, which have attracted the world's attention [8]. To protect and develop traditional villages, China proposed the "Rural Revitalization Strategy" in 2017 [9].

Traditional villages are also essential carriers of Chinese civilization's local memory and historical culture. They are also the concentrated expression of the inheritance of rural

charm and value under the background of rural revitalization. Protecting and inheriting traditional villages are to preserve and inherit historical culture. Therefore, how to protect and inherit traditional villages has become a significant issue in today's academic circles [10,11]. Law is the foundation of understanding things, and space is the carrier of cultural inheritance. In rural planning, we should firmly grasp the evolutionary law of traditional villages and explore the space features of traditional villages. Only in this way can we lay a solid foundation for revitalizing traditional villages. Aiming to explore the objective laws of the growth, development, and layout of traditional villages more accurately, we introduce the "Space Gene" concept to deconstruct and quantify traditional villages. Moreover, by shifting from the "built form" research from the perspective of material space to the "deep structure" research behind it, we explored a new way to re-examine traditional village space research. To test the theory, we take Shibadong Village as an example in the Xiangxi Autonomous Prefecture, Hunan Province.

The rest of the article is as follows. We first outline the literature on traditional village forms. Next, a quantitative model of traditional village forms is proposed. Then, we introduce the study area and validate the model and, finally, discuss and summarize the results.

## 2. Literature Review

As a branch of urban morphology research and development, the space cognition of traditional village morphology is not a new topic. As early as 1841, J. G. Kohl analyzed the relationship between settlement form and topography [12]. Moreover, in 1895 A. Meitzen conducted a field study of agricultural settlements in northern Germany and classified these settlement forms [13]. However, research in this period focused on the description of the phenomenon. After the 1960s, as computer technology developed, a combination of quantitative and qualitative methods was gradually adopted, and human behavior and psychological factors were introduced into the studies of rural settlements [14–17]. These methods have been further developed in recent years; for example, in 2003, the notion of Cultural Landscape Genes of Traditional Settlements (CLCTS) was first put forward by PL Liu et al., and they identified and classified the space structure of traditional villages [18]. Zui Hu et al. (2015) used semiotics to visualize the CLCTS and established a database [19]. In order to quantify changes in traditional rural landscape markers at the farm scale, Daniele Torreggiani proposed the TRuLAn method (Traditional Rural Landscape Analysis) in 2014 [20]. Xi Jianchao et al. used the PRA (Participatory Rural Assessment) method, GIS technology, and high-resolution remote sensing images to analyze the evolution of tourism-induced village spatial morphology [21]. Some scholars have studied the individual residents [22,23], distribution characteristics [24–26], land use [7,27], etc., in traditional villages. Moreover, some scholars have studied and discussed the influencing factors of forming traditional villages, including the natural geographical environment, economic conditions, policy mechanisms, and so on [28–31].

However, to achieve the inheritance of the traditional village context, it is not enough to only study traditional villages' historical forms and symbols. We need to dig into the regional space element combination mode and the continuation logic of its internal texture. Using the "Space Gene" concept to analyze traditional villages can effectively make up for this defect.

The "Space Gene" originates from the theory of urban complex systems [32–34]. It has an individual and relatively stable space combination mode. It is not only the product of a profound fusion and accommodation of ecological environment, urban space, culture, and history but also carries the individual information about different regions; forms the identity of urban characteristics; and plays a role in maintaining the harmonious relationship among the city, nature, and history [35]. When analyzing specific spaces, we can achieve a directional shift from universal to local in planning and design based on the space gene to avoid adopting a unified model for urban construction in the face of vastly different cities [36].

Cities initially originated from villages, so we put forward the traditional village space gene concept. It is an individual and a stable space combination mode that contains the fundamental law that traditional villages can be stably inherited. It constitutes traditional villages' space pattern, form, and space structure. It serves as the carrier of cultural heritage and the fundamental inheritance unit that enables traditional villages to develop continuously. Moreover, we further constructed the Quantitative Inheritance System Model of Traditional Village Space Genes, which can reveal the logic behind the traditional village material form formation. By this model, we can conduct a more comprehensive quantitative analysis of the traditional village form.

## 3. The Quantitative Inheritance System Model of Traditional Village Space Genes

Based on the research background above and related literature analysis, we propose a comprehensive framework of a Quantitative Inheritance System Model for Traditional Village Forms Based on Space Genes, as shown in Figure 1. Three features of the space gene inspire it: hierarchies—space genes exist in cities, neighborhoods, streets, courtyards, and other levels; self-organization—space genes are replicated and expressed in the process of urban self-organization, and a relatively stable space combination pattern is formed through mutation and selection; opening—space genes generate new space information due to the opening of the city, which can promote optimization, generate a new space stability model, and finally realizes the evolution of space forms at different levels [36]. In addition, we also draw on the genetic principle of biological genes [37]. Space genes are also information carriers that control the space forms (features) of materials and store key information on the evolution and development of traditional village space forms.

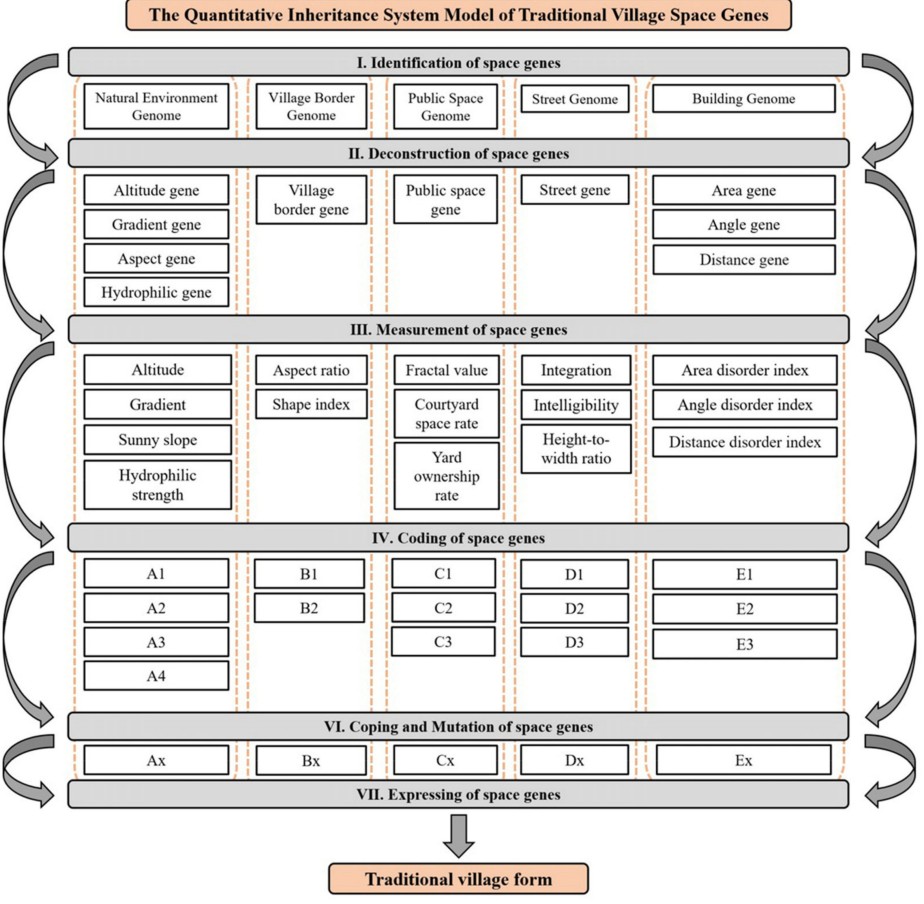

**Figure 1.** The general framework of the proposed research model. Source: Author.

In addition, space genes carry the spatial information of the interactive evolution of "urban space-natural environment-social humanities" and play a vital role in the inheritance of urban space context [36]. Therefore, the interpretation of spatial information such as traditional village space, natural environment, and social culture is the key to deconstructing and measuring the space genes of traditional villages. Our proposed model consists of six steps:

At first, the traditional village space is identified from outside to inside, including five genomes of the natural environment, village border, public space, streets, and buildings.

The second step is to deconstruct the genome. For example, the natural environment genome is deconstructed into altitude, gradient, aspect, and hydrophilic genes; the village border genome is deconstructed into village border genes, etc.

The third step is to measure the genes. The precise quantification of each gene through the corresponding indicators can reflect the natural environment, space texture, and internal social order of traditional villages.

The fourth, fifth, and sixth steps include gene coding, gene duplication and mutation, and gene expression.

The above six steps allow all space genes to be translated into traditional village forms. The process reveals the logic behind forming the material form of traditional villages.

Therefore, introducing the theory of "Space Gene" in the research of traditional villages to further understand and analyze the traditional village space genes will be of great significance to realize the inheritance of the traditional village space context. In the process of rural revitalization, it is helpful to achieve the possibility of a win–win situation between the historical inheritance of traditional villages and contemporary development. This research has a specific guiding significance and reference value for rural planners and managers.

## 4. Materials and Methods

### 4.1. Study Area

The Shibadong Village is located in central China in the Xiangxi Tujia and Miao Autonomous Prefecture, Hunan Province. It is the birthplace of the theory of "targeted poverty alleviation". Additionally, it is 34 km from Huayuan County and 38 km from Jishou City, the capital of Xiangxi Prefecture. It is the administrative village of Shuanglong Town. There are four natural camps: Feichong Camp, Dangrong Camp, Zhuzi Camp, and Lizi Camp. Moreover, it belongs to the pure Miao inhabited area, as shown in Figure 2. The western Hunan region is one of the typical multi-ethnic agglomerations in China with relatively backward transportation, an underdeveloped economy, complex mountainous terrain, and relatively fragile ecology. Furthermore, many traditional villages with national features are preserved here. Villages reflect unique regional cultural landscapes due to the influence of landscape elements such as terrain conditions, ancient traffic routes, ethnic minorities' politics and militaries, patriarchal concepts, religious rituals, folk customs, and cultural psychology. As one of the traditional villages in Western Hunan, Shibadong Village has a long history of farming terrace culture, profound Miao culture, unique Miao food culture, strong Miao customs, and well-preserved original ecological culture. Moreover, there are Miao buildings with regional characteristics, such as stilted buildings, towers, fire ponds, etc.

### 4.2. Data Sources

The research data includes village-level administrative divisions, rivers and roads, Digital Elevation Model data, satellite images, point data of traditional villages, photos of field surveys, and manually supplemented building and courtyard outline data based on satellite images. The above data are all vector data.

The data on administrative divisions, rivers, and roads at the village level were provided by the High Definition Database of Hunan Provincial Architectural Design Institute, which is beneficial to the definition of the study area. The terrain data is derived

from the Geospatial Data Cloud (http://www.gscloud.cn/ (accessed on 7 June 2022)). Satellite image data came from Google Earth with an accuracy of 18, facilitating the later addition of vector data such as building outlines and courtyard outlines in the study area. To ensure the accuracy of the data in the study area, the site data downloaded from the Resource and Environmental Science and Data Center (http://www.resdc.cn/ (accessed on 7 June 2022)) was used. At the same time, in October 2021, the actual survey and photographing of the research area were carried out. In addition, all vector data about the study area were accurately corrected in ArcGIS 10.3 to ensure data consistency and validity.

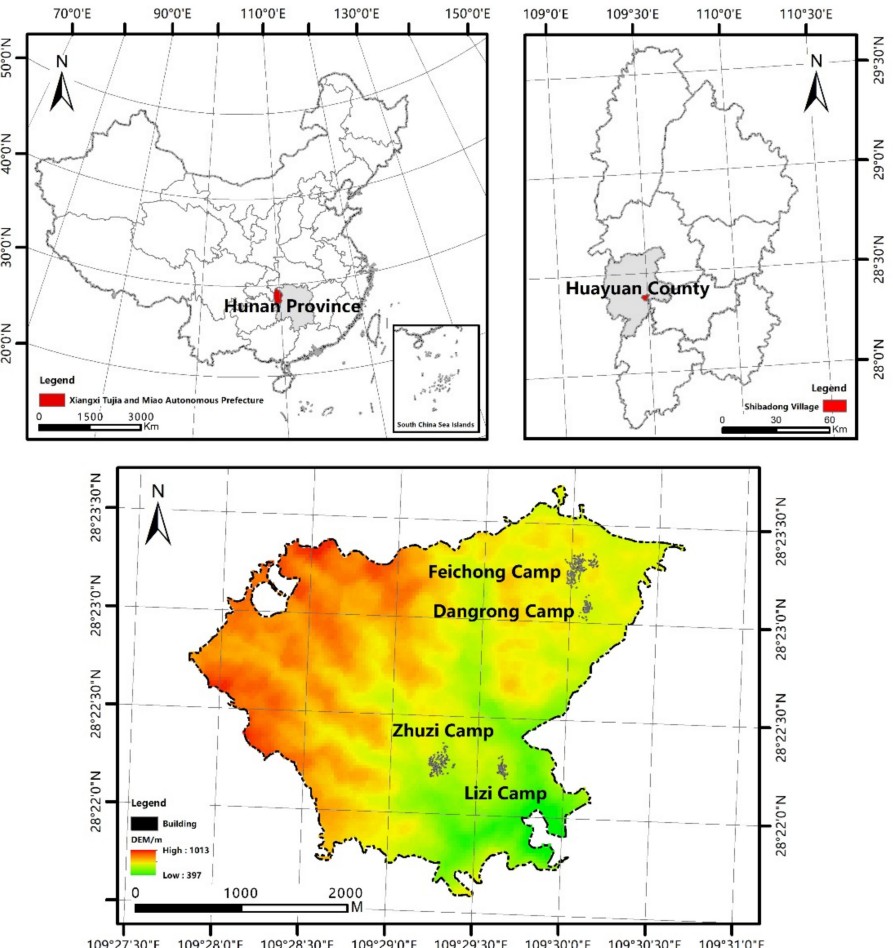

**Figure 2.** Geographical location of Shibadong Village. Source: Author.

### 4.3. Methods

According to The Quantitative System Model of Traditional Village Space Genes constructed in this paper, we analyzed the overall morphology of traditional villages to explore the mechanisms and laws behind the abstract forms. The system model comprehensively adopts interdisciplinary methods such as urban geography, landscape ecology, fractal geometry, space syntax, computer programming, and statistics from morphological diagrams to quantitative statistics. The following are the research methods used in the system model. Additionally, all data calculations and analyses were performed in ArcGIS and Rhino.

#### 4.3.1. Shape Index

To deconstruct the border form of traditional villages more scientifically, we set the virtual border of the scope from the macro, meso, and micro levels. Previous studies have shown that 100 m is the highest limit of social vision and ability to identify specific individuals, but it is not clear who they are or what they are doing. At 30 m, people's

facial features, hairstyles, and ages can be seen, and people you do not meet often can be recognized. Seven meters is the demarcation point between the "close phase" range (3~7 m) and the "far direction" range (7~20 m) in the "mutual knowledge domain" [38–41]. However, it is not easy to connect to form a complete outer border shape when using 7 m as the maximum span in the study area. Therefore, we set three scale levels of 100 m, 30 m, and 12 m as the maximum distance that the virtual border can span and define it as the large border, the middle border, and the small border.

The shape index is a mathematical index extensively applied in landscape ecology. Additionally, it is based on the shape index of a compact shape (circle, square, rectangle, or other regular polygons, etc., as needed) as a reference standard. For example, if a circle is used as a reference, we take the ratio of the perimeter of the figure to the circle of equal area. Then, we obtain the circularity based on the perimeter of the image outline. The circularity can reflect the degree of deviation between the shape of the figure and the circle of equal area, also called "shape deviation" [42]. To measure the border of traditional villages more accurately, we select an ellipse of equal area and an equal aspect ratio as a reference. The specific calculation method is shown in Equation (1):

$$\lambda = \frac{a}{b}$$
$$S = \frac{P}{\left(1.5\lambda - \sqrt{\lambda} + 1.5\right)} \sqrt{\frac{\lambda}{A\pi}} \tag{1}$$

where $a$ and $b$ are the length (m) and width (m) of the minimum circumscribed rectangle of the traditional village form border, $P$ is the perimeter of the traditional village border (m), $A$ is the area of the traditional village border (m$^2$), and $\lambda$ is the aspect ratio of the traditional village border.

### 4.3.2. Fractal Dimension

Fractal theory was proposed by Mandelbrot in 1975 [43]. This theory mainly researches the space features and temporal evolution of earth phenomena [44]. Fractal theory is mainly applied to urban research issues, including morphological borders, structures, growth, transportation, and evolution mechanisms [45–48]. There are three calculation methods of fractal dimension: area-perimeter relation, box-counting method, and area–radius relation. In some cases, the area–perimeter relation and box-counting method are the same, but the area–radius relation helps characterize urban growth and dynamics from a dynamic perspective [49]. For the study of traditional village forms, since we do not involve the study of the dynamic growth of traditional villages, the relatively simple first method is selected. The calculation method is as Equation (2):

$$D = \frac{2\lg(\frac{P}{4})}{\lg(A)} \tag{2}$$

Among them, $D$ represents the fractal dimension, $P$ is the perimeter (m), and $A$ is the patch area (m$^2$).

The theoretical value of $D$ is 1.0~2.0. The shape represented by 1.0 is the simplest square pattern patch, while the shape represented by 2.0 is the most complex pattern patch with an area perimeter.

### 4.3.3. Integration and Intelligibility

Integration can reflect the degree of aggregation and dispersion of unit space with all other spaces in the system [50,51]. As Equation (3), we have:

$$I = \frac{2(MD - 1)}{n - 2} \tag{3}$$

In Equation (3), $n$ means the total number of axes or nodes in the space system, while *MD* means the average depth. Its calculation method of *MD* is as Equation (4):

$$MD_i = \sum_{i-1}^{n} \frac{d_{ij}}{(n-1)}$$
$$D_n = \frac{2\{n[\log_2(\frac{n+2}{3}-1)+1]\}}{(n-1)(n-2)} \tag{4}$$

Intelligibility is the ratio of connectivity to global integration, which is one's ability in a local position to perceive the overall space [51,52]. The connectivity represents the number of other unit spaces in the system that intersect the i-th unit space. Corresponding to the axis graph, the connectivity represents the total number of other streets that intersect the specified street i. The calculation method is as follows in Equation (5):

$$C_i = k$$
$$R^2 = \frac{\left[\sum_{1}^{n}\left(C_i-\overline{C}\right)\left(I_{(n)}-I'_{(n)}\right)\right]^2}{\sum_{1}^{n}\left(C_i-\overline{C}\right)^2 \sum_{1}^{n}\left(I_{(n)}-I'_{(n)}\right)^2} \tag{5}$$

In Equation (5), $\overline{C}$ is the mean of the space connectivity values of all units, $I_{(n)}$ is the global integration degree value, and $I'_{(n)}$ is the global average integration degree.

### 4.3.4. Directional Vector

Traditional villages are composed of a series of building units. They are determined by the three primary vectors of their area size, square angle, and distance from other building units. In aggregating into an overall network, they construct the overall spatial structure of traditional villages. Therefore, the above three primary vectors can well deconstruct architectural genes and then interpret the spatial form of traditional villages. Their calculation formulas are as follows:

$$\sigma = \frac{\sum_{1}^{n} S_n}{n}$$
$$A = \sigma / \sqrt{\frac{\sum_{1}^{n}(S_n-\sigma)}{n}} \tag{6}$$

In Equation (6), $\sigma$ means the average value of the area of the single building (m), $S$ means the acreage of the single building (m$^2$), and $A$ means the acreage disorder index of the single building.

$$\alpha = \arccos\frac{a_1 b_1 + a_2 b_2}{\sqrt{a_1^2+b_1^2} \times \sqrt{a_2^2+b_2^2}}$$
$$B = \frac{\sum_{1}^{n}(|\alpha_n-45|)/45}{n} \tag{7}$$

In Equation (7): $B$ represents the directional ordinal value, which is 0~1. The larger $B$ is, the higher the consistency of house orientation and the better the village order is; n represents the number of node angles; and a1, b1, a2, and b2 represent the coordinates of the two endpoints of the line segment connecting the center lines of the two-building monomers, while $\alpha$ represents the angular difference between axes with spatial associations.

$$\beta = \frac{\sum_{1}^{n} L_n}{n}$$
$$C = \beta / \sqrt{\frac{\sum_{1}^{n}(L_n-\beta)}{n}} \tag{8}$$

In Equation (8), $\beta$ is the average distance between two building units (m), $L$ is the distance between two building units (m), and $C$ is the distance disorder index of the building units.

## 5. Results

### 5.1. Natural Environment Genome

This part shows the calculation results of the altitude, slope, aspect, and hydrophilic strength of Shibadong Village.

#### 5.1.1. Altitude Gene

According to the analysis results of the DEM, the minimum altitude of the study area is 397 m, and the highest altitude is 1013 m. Based on the terrain features of the study area, it is further subdivided into six intervals by using the natural discontinuous point splitting method in ArcGIS, including <397 m, 397~500 m, 500~600 m, 600~700 m, 700~800 m, and 800~1013 m. After superimposing it with the residential data, the spatial distribution of residential buildings at different altitudes was obtained to reflect the altitude distribution features of the residences in the study area, as shown in Figure 3.

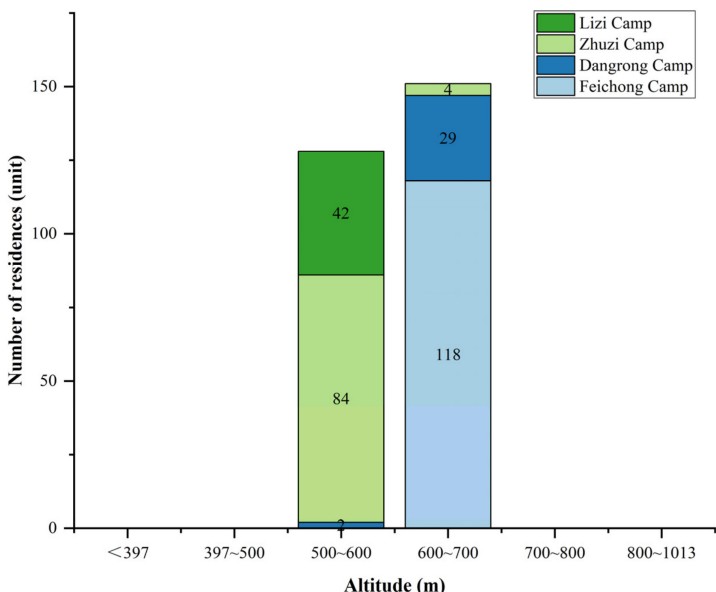

**Figure 3.** The distribution of residences in Shibadong Village at different altitudes. Source: Author.

It can be seen from Figure 3 that the residences in the four camps are concentrated in the area of 500–800 m above sea level, and no residents are living below 500 m or above 700 m above sea level. According to the altitude data of the study area, its residential dwellings are all distributed in the mid-altitude section of the area.

#### 5.1.2. Gradient Gene

According to the Slope Classification of the International Geographical Union Commission on Geomorphic Survey and Geomorphometric Mapping for the Application of Detailed Topographic Maps, the slope is divided into five intervals, including $\leq 2°$, $2°\sim6°$, $6°\sim15°$, $15°\sim25°$, and $>25°$. A slope $\leq 2°$ is considered a flat ground, a slope of $2°\sim6°$ is a gentle slope, a slope is of $6°\sim15°$, a moderately steep slope is $15°\sim25°$, and a steep slope is above $25°$. After overlay analysis, we can obtain the spatial distribution of residential buildings on different slopes, as shown in Figure 4.

It can be seen that about 57% of the residents in Feichong Camp are located in the slope area of 0~15°; about 65% in Dangrong Camp, about 49% in Zhuzi Camp, and only 20% in Lizi Camp are here. Within the slope range, as the slope exceeds 15°, the development of land resources is complex, and soil erosion is severe, but many residents still live in this

environment. In general, Feichong Camp, Dangrong Camp, and Lizi Camp are mainly located in the range of 2~25°, and Zhuzi Camp is mainly distributed in the range of 6~25°.

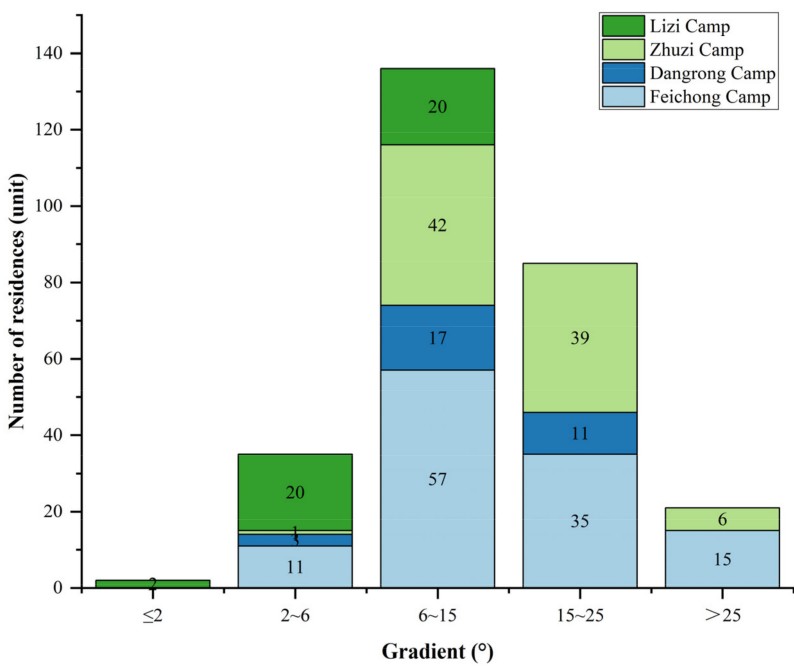

**Figure 4.** The distribution of residences in Shibadong Village on different gradients. Source: Author.

### 5.1.3. Aspect Gene

In this paper, the slope aspect is divided into five categories, including shady slopes (0~45°, 315°~360°), semi-shady slopes (45°~135°), sunny slopes (135°~225°), and semi-sunny slopes (225°~315°). The results are shown in Figure 5. In addition, without considering the sunshine time, we use the sunny slope rate to measure the proportion of the number of residences on the sunny slope. For residential houses located on semi-shady and semi-sunny slopes, we take half of the number, add the number of residential houses on the sunny slope, and finally calculate the sunny slope rate. After calculation, it was found that the sunny slope rate of Feichong Camp is 0.56, that of Dangrong Camp is 0.58, that of Zhuzi Camp is 0.73, and that of Lizi Camp is 0.71.

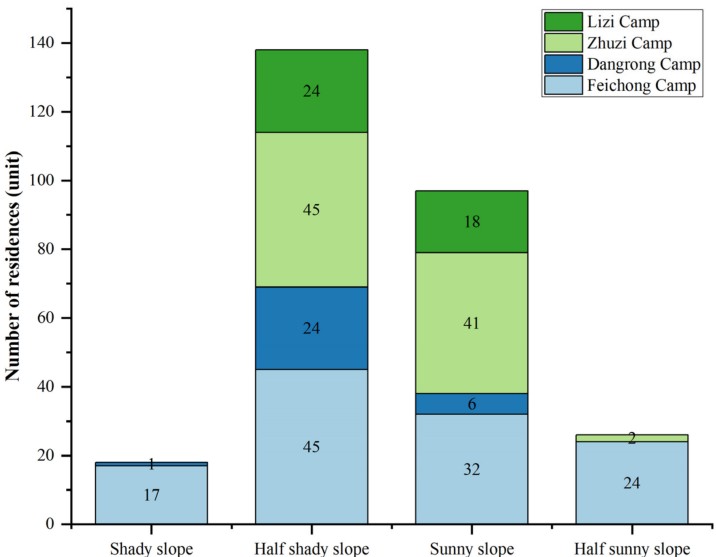

**Figure 5.** The distribution of residences in Shibadong Village on different aspects. Source: Author.

### 5.1.4. Hydrophilic Gene

The hydrophilic analysis adopts the distance analysis method. The shortest distance from the building center point to the river border is the hydrophilic index. The shorter the distance, the stronger the hydrophilicity. The shorter the distance, the stronger the hydrophilicity. For each camp, we selected the average hydrophilicity as the hydrophilicity of the whole camp. After analysis, the obtained results are shown in Figure 6. The hydrophilicities of Feichong Camp and Dangrong Camp are very weak, at 907 m and 963 m, respectively; Zhuzi Camp is the strongest at 147 m; and Lizi Camp is slightly weaker at 309 m.

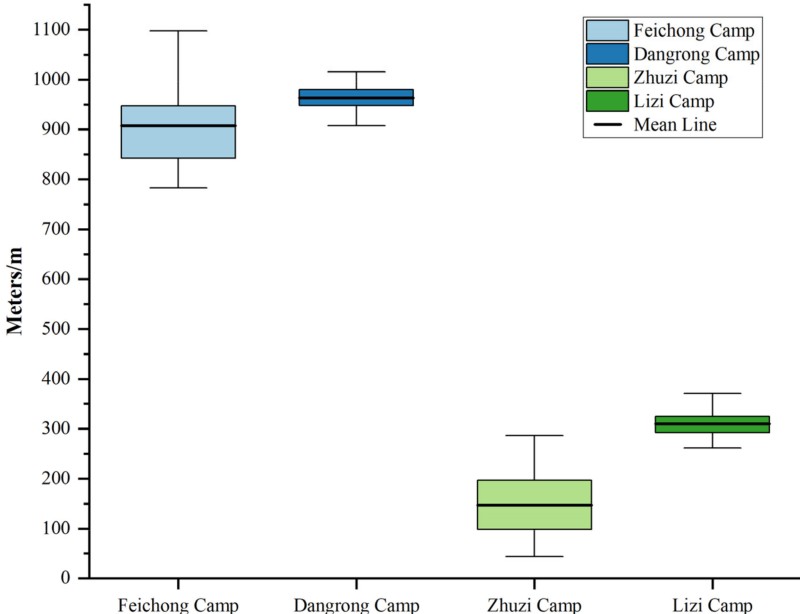

**Figure 6.** Hydrophilic strength of the four camps in Shibadong Village. Source: Author.

### 5.2. Village Border Genome

This part mainly shows the calculation results of aspect ratio and shape index. We have drawn the big, medium, and small borders of the four camps, as shown in Figure 7. According to the calculation method of the shape index selected in this paper, the morphological features and related indicators of the four camps' borders are mathematically calculated. In addition, to obtain a comprehensive index as an example, we have weighted the three shape indices. The data are shown in Table 1.

**Table 1.** Morphological indexes of a border.

|  | λ | S1 | S2 | S3 | S |
|---|---|---|---|---|---|
| Feichong Camp | 1.02 | 1.23 | 2.41 | 5.18 | 2.27 |
| Dangrong Camp | 2.31 | 1.16 | 1.45 | 3.14 | 1.53 |
| Zhuzi Camp | 1.35 | 1.56 | 1.92 | 5.16 | 1.99 |
| Lizi Camp | 2.21 | 1.07 | 1.49 | 2.82 | 1.48 |

λ: Aspect ratio, S1: Shape index for small borders, S2: Shape index for middle borders, S3: Shape index for big borders, S: Weighted shape index.

According to Table 2 [38], it can be clearly distinguished what type of village the four camps are. Feichong Camp is a clump-prone, finger-like village; Dangrong Camp is a banded village; Zhuzi Camp is a clump village; and Lizi Camp is a banded village.

### 5.3. Public Space Genome

This part calculates the parameters of public space and courtyard space. We use this as the scope to convert the figure to the bottom according to the closed figure of the outer

border set for the traditional village. The part between the building entity and the border is the external space of the village, which includes the public space and the courtyard space [38]. According to Figure 7, the middle border can more accurately reflect the shape of traditional villages. In order to make the public space form a pattern, the width of the outer edge of the public space of the traditional village increases by 2.5 m [38,41], as shown in Figure 8.

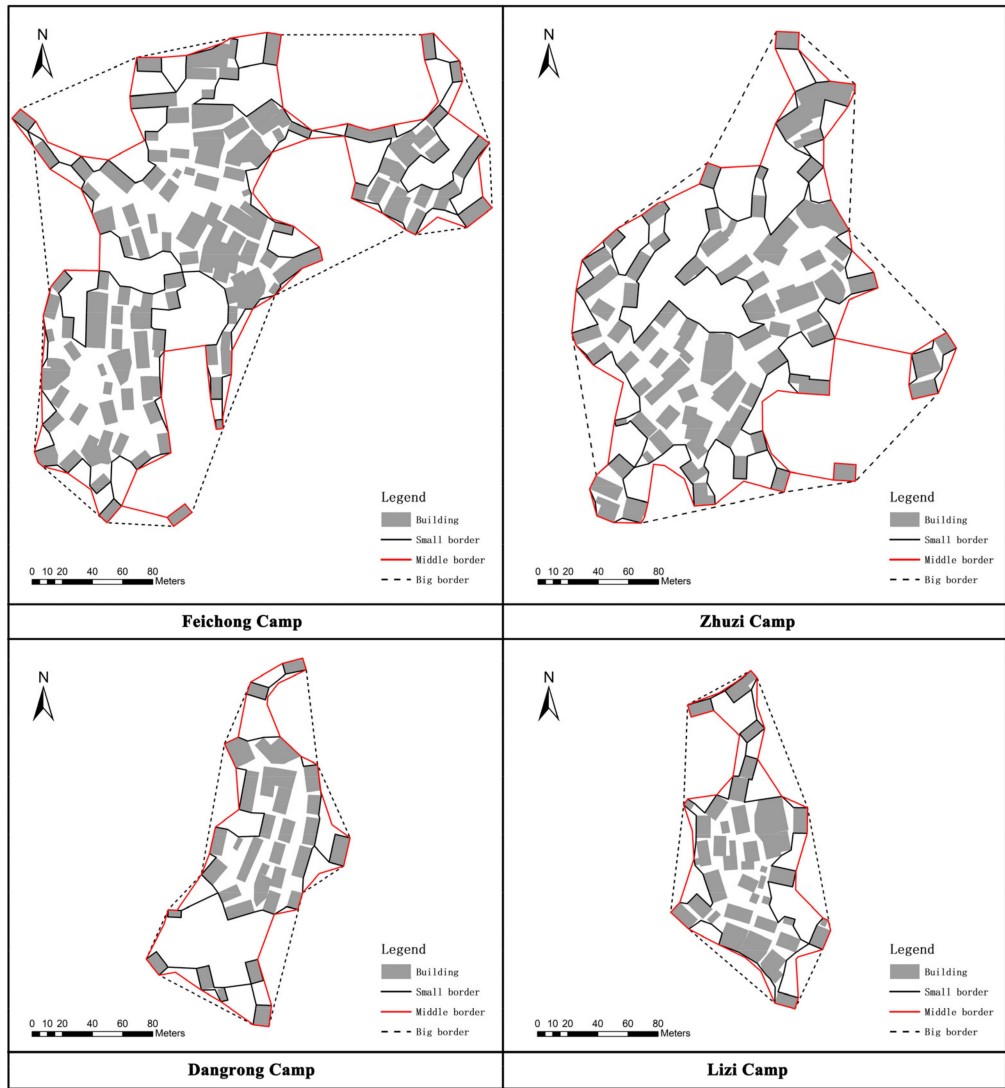

**Figure 7.** The illustrations of the village border genes in Shibadong Village. Source: Author.

**Table 2.** Traditional village form classification.

| S | λ | Type |
|---|---|---|
| S ≥ 2 | Λ < 1.5 | Clump-prone, finger-like village |
| | 1.5 ≤ λ < 2 | Finger-like village with no clear inclination |
| | Λ ≥ 2 | Band-prone, finger-like village |
| S < 2 | Λ < 1.5 | Clump village |
| | 1.5 ≤ λ < 2 | Band-prone clump village |
| | Λ ≥ 2 | Banded village |

S: Shape index λ: Aspect ratio.

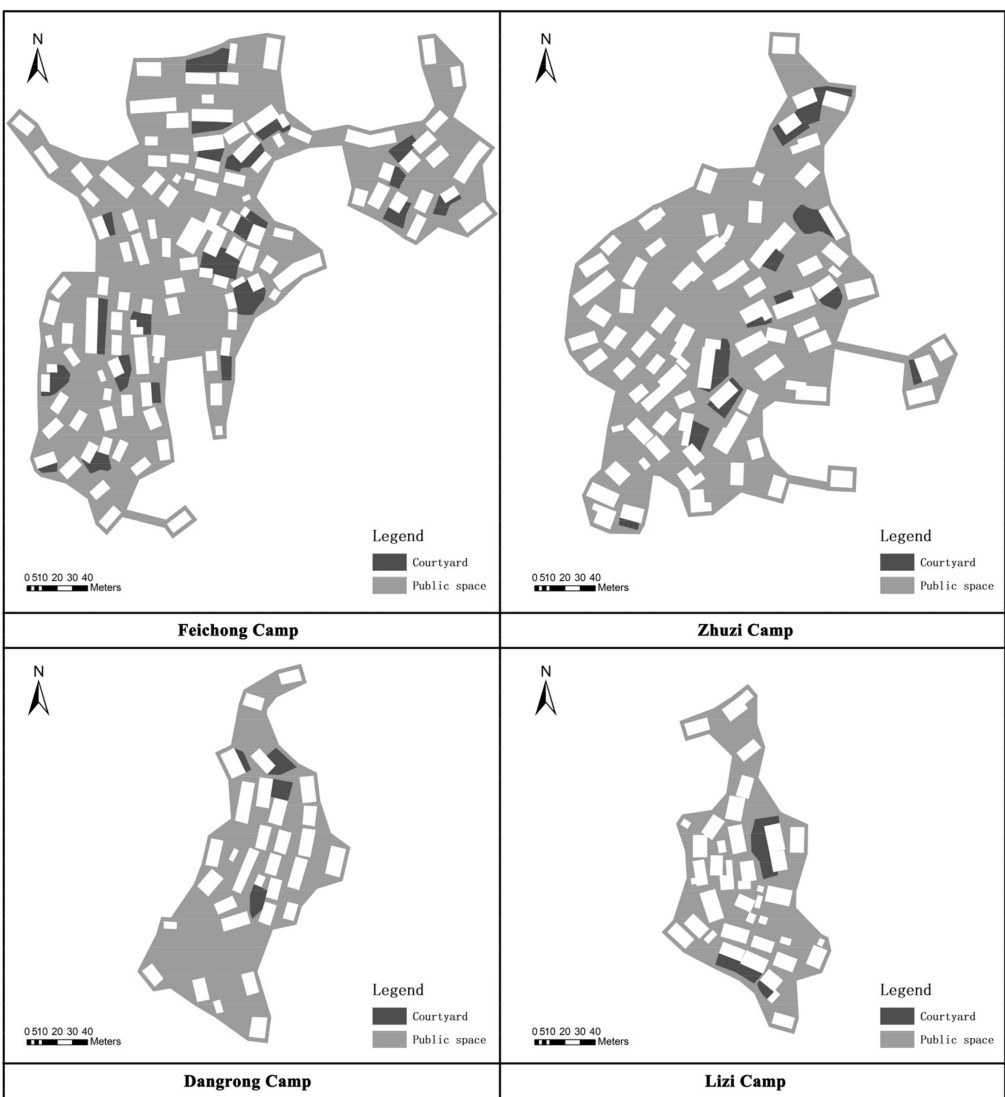

**Figure 8.** The illustrations of the public space genes in Shibadong Village. Source: Author.

The scientific significance of fractal dimension value is to characterize the complexity and fragmentation of the patch. The more complex and broken the patch, the higher the fractal dimension value. When this fractal dimension value gets higher, the public space of a particular area is more fragmented. There are more interface enclosures, and the sense of spatial experience will become more diverse and complex as well as more prosperous. Therefore, the higher the fractal dimension, the stronger the structure of its agglomeration and vice versa. In addition, the area defined by the outer contour of the combination of the single building and the courtyard is used as the base, and the statistics of the space ratio of the courtyard are carried out. According to the number of building units with courtyards, the statistics of the courtyard ownership rate are carried out. After calculation, the obtained results are shown in Table 3. The fractal dimension value, courtyard space rate, and courtyard ownership rate of Feichong Camp are the highest, indicating a public space with the highest agglomeration degree. Dangrong Camp is the lowest, and Zhuzi Camp and Lizi Camp are in between.

### 5.4. Street Genome

According to the spatial texture map of Shibadong Village and the actual survey results, axes are used to represent the network skeleton of Shibadong Village's streets and lanes, forming a spatial axis map of Shibadong Village. Additionally, we imported it into

Dethmap to calculate the integration and intelligibility and generate the regression analysis graph of the connection value axis graph, the global integration degree axis graph, and the comprehensibility (Figure 9).

**Table 3.** Indicators of the public space gene.

|  | D | Courtyard Space Rate | Yard Ownership Rate |
|---|---|---|---|
| Feichong Camp | 1.46 | 0.22 | 0.2 |
| Dangrong Camp | 1.37 | 0.13 | 0.13 |
| Zhuzi Camp | 1.42 | 0.15 | 0.15 |
| Lizi Camp | 1.43 | 0.14 | 0.1 |

D: Fractal dimension.

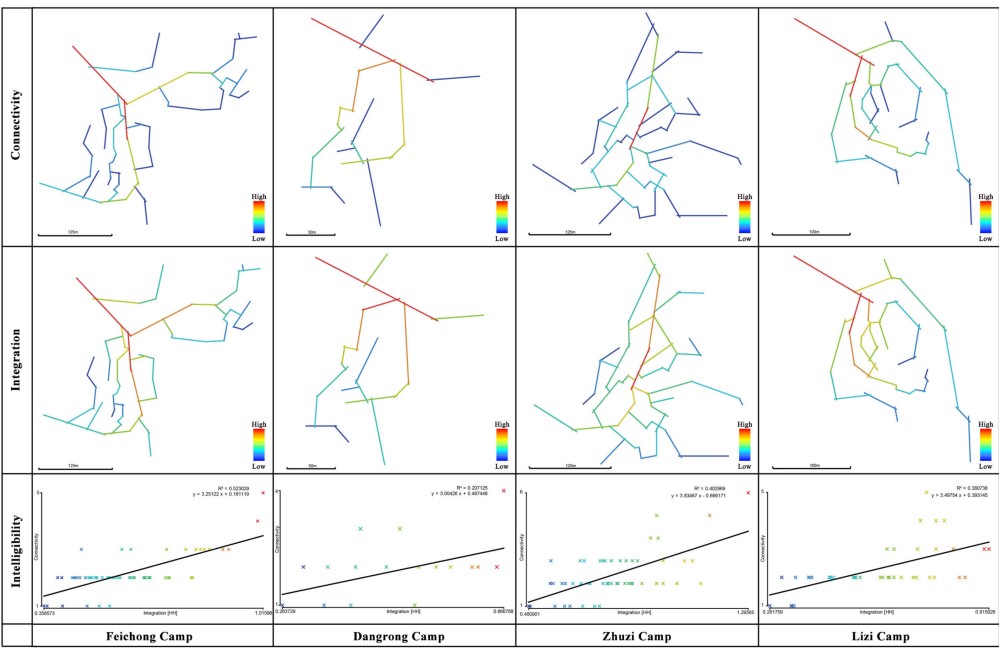

**Figure 9.** The integration and intelligibility of the Shibadong Village. Source: Author.

In addition, according to the photos of the on-site investigation, we draw the typical street profiles of the four camps (Figure 10). The streets of Feichong Camp and Lizi Camp are relatively spacious, and the foundation heights of houses in Lizi Camp vary greatly. The height-to-width ratios of their streets are 0.81 and 0.6, respectively. The streets of Zhuzi Camp and Dangrong Camp are very compact, and the height-to-width ratios of the streets and alleys are 1.75. However, the house foundation of Zhuzi Camp has a slight height difference.

After calculation, various indicators of the Street genes of the four camps in the Shibadong Village can be obtained in Table 4. The higher the integration, the higher the spatial accessibility of the region and the intelligibility, indicating that the local space structure of the region is beneficial to establishing the perception of overall space. Therefore, according to the calculation results, Zhuzi Camp has the highest accessibility, Dangrong Camp has the lowest, and Feichong Camp and Lizi Camp are in the middle; meanwhile, Feichong Camp has the most complete spatial structure, Dangrong Camp has the worst spatial structure, and the spatial structure of Zhuzi Camp and Lizi Camp are in the middle.

### 5.5. Building Genome

This part of the calculation was performed in Rhino and shows the calculation results of building area, angle, and distance. According to Equations (6)–(8), the area disorder index, angle disorder index, and distance disorder index of the building monomer in Shibadong Village were calculated, as shown in Table 5.

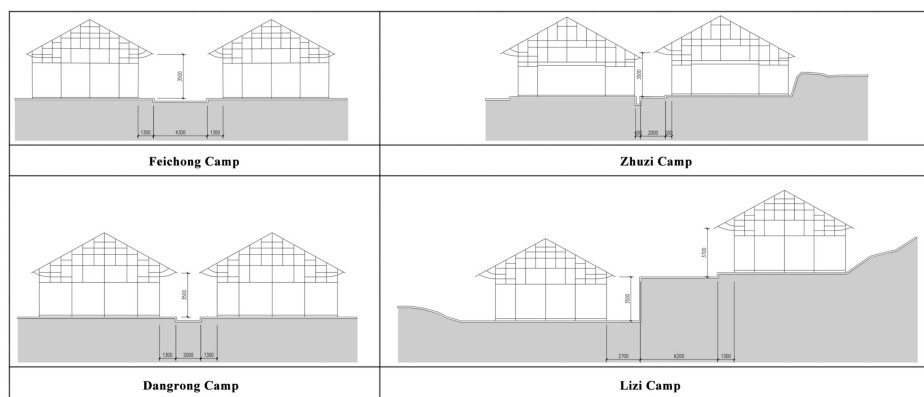

**Figure 10.** Sectional drawing of streets in Shibadong Village. Source: Author.

**Table 4.** Indicators of the Street gene.

|  | Integration | Intelligibility | Height-to-Width Ratio |
|---|---|---|---|
| Feichong Camp | 0.61 | 0.52 | 0.81 |
| Dangrong Camp | 0.47 | 0.21 | 1.75 |
| Zhuzi Camp | 0.8 | 0.4 | 1.75 |
| Lizi Camp | 0.54 | 0.38 | 0.6 |

**Table 5.** Indicators of the Area gene, Angle gene, and Distance gene.

|  | σ | A | B | β | C |
|---|---|---|---|---|---|
| Feichong Camp | 106.26 | 0.51 | 0.41 | 40.24 | 0.55 |
| Dangrong Camp | 128.58 | 0.37 | 0.33 | 39.77 | 0.53 |
| Zhuzi Camp | 127.12 | 0.36 | 0.6 | 42 | 0.51 |
| Lizi Camp | 105.04 | 0.52 | 0.42 | 30.97 | 0.58 |

σ: Average value of the area, A: Area disorder index B: Angle disorder index, β: Average distance, C: Distance disorder index.

As shown in Figure 11, it can reflect the differences in the distribution of the area, angle, and distance of the individual buildings in the four camps. The consistency of the area of Feichong Camp and Lizi Camp is relatively high. Although the areas of Dangrong Camp and Zhuzi Camp are large, the consistency is weak; the angular consistency of Zhuzi Camp is the best, but Dangrong Camp is the worst; however, the consistency of the distances between the building units in the four camps is almost the same.

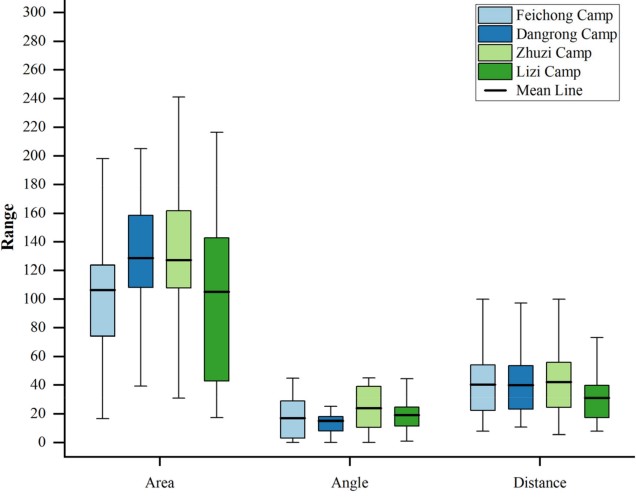

**Figure 11.** The distribution of buildings in Shibadong Village at area, angle, and distance. Source: Author.

## 6. Discussion and Conclusions

### 6.1. Discussion

The formation and development of the spatial form of traditional villages are closely related to many factors, including the natural environment, traditional culture, and social policies.

(1) The natural environment is the background foundation of traditional village forms. Western Hunan is a mountainous terrain with numerous rivers. This unique natural environment dramatically affects the space texture of traditional villages. Combined with the case study, it reflected that traditional villages are different in the plane and vertical senses. However, the development and construction of traditional villages are backed by mountains, facing open fields and surrounded by hills on four or three sides. Because of the humid and rainy climate in Western Hunan, most traditional villages are not located by rivers, and the mountain spring water is enough for people to live. These have led to the various forms of traditional villages, showing various forms such as lumps, bands, and fingers.

(2) Traditional culture is the spiritual connotation of traditional village forms. Based on the influence of clan culture, surname villages are often independent and complete living circles, forming a unique model of living together. The internal agglomeration of such traditional villages is very strong, which leads to the fact that although some dwellings are located in harsh environments, the buildings of the same ethnic group are closely distributed, enabling the villages to develop in a stable and orderly manner.

(3) Social economy is the motive power for developing traditional village forms. From ancient times to the present, the terrain of Western Hunan has been complicated, and the traffic connection with the outside world has been greatly hindered, which has led to the social and economic backwardness compared with other regions, making it a place for ethnic minorities in China to escape from war and disasters. The region has a suitable climate and fertile land, which provides a specific material basis for developing traditional villages. Therefore, Western Hunan is one of the regions with the complete preservation of traditional villages in China.

At present, most studies on the space features of traditional villages are due to a certain factor or a certain calculation method to study the morphological features of traditional villages [53,54] or to research the formation process of villages from the perspective of spatial semiotics [19,55]. However, they often ignore exploring the formation mechanism behind the material space. In addition, there are also studies on the impact of land use on rural landscape patterns [56,57]. However, research from a macro perspective is challenging to guide planning and design at the individual level of villages. In this paper, it can not only systematically and comprehensively analyze the form of traditional villages and shows the laws behind the formation of their material forms but also provide scientific and practical guidance for the planning and design of traditional villages.

In addition, the highlights are below. First, the theory of space genes is introduced into the study of traditional village forms, and the concept of traditional village space genes is defined. Second, drawing on the mechanism of action of genes in biology, a Quantitative Inheritance System Model of Traditional Village Space Genes is proposed. Third, it reveals the logic behind the formation of the material form of traditional villages. Fourth, research has proved that it is possible to accurately quantify the traditional village form and explore the laws behind it.

However, any analysis is partial and one-sided. We quest for the morphological features of traditional villages in the natural environment, village borders, public spaces, streets, and architectural order. There are many other perspectives, such as the historical elements of traditional villages, landscape viewing corridors, architectural forms, colors, etc. Moreover, we did not compare the relationship between different traditional village forms in terms of region, economy, or culture. In addition, we only used one case to study, so the research in this paper must have certain limitations. We will increase the type and number of traditional villages in follow-up research.

*6.2. Conclusions*

Table 6 shows the quantitative values of each space gene for the four stockades in Shibadong Village. Even though they belong to the same village and one ethnic group, there are still subtle differences within them. Leading conclusions:

(1) The four camps are all located at mid-altitude. Nearly 40% of the residents of Feichong Camp, Dangrong Camp, and Zhuzi Camp live in places with a slope of more than 15°. More than 70% of Zhuzi Camp and Lizi Camp residents live on sunny slopes. Only Zhuzi Camp is located on the edge of the river.

(2) The shapes of village borders are not the same. Feichong Camp is a finger-shaped village with a tendency to cluster, Dangrong Camp and Lizi Camp are both band-shaped villages, and Zhuzi Camp is a cluster-shaped village.

(3) Feichong Camp has the most robust public space agglomeration; its courtyard rate and courtyard ownership rate are also the highest, and the other three stockades are similar.

(4) The intelligibility of the public space of Feichong Camp is the highest, the integration degree of Zhuzi Camp is high, and the height-to-width ratio of streets and alleys between Dangrong Camp and Zhuzi Camp is the largest.

(5) The area disorder index of Feichong Camp and Lizi Camp is the largest, and the angle disorder index of Zhuzi Camp is the largest. However, the distance disorder index of the four villages is almost the same. Based on the Quantitative Inheritance System Model of Traditional Village Space Genes, the results show that we revealed the logic behind the formation of the traditional village material form and realized the systematic quantification of the traditional village form. On this basis, we can better protect the built cultural heritage. Rural planning and design can also respond promptly to the development of modern society to achieve true cultural heritage and protect world cultural diversity.

**Table 6.** Gene coding table of Shibadong Village.

| Genome | A | | | | B | | C | | | D | | | E | | |
|---|---|---|---|---|---|---|---|---|---|---|---|---|---|---|---|
| Gene | A1 | A2 | A3 | A4 | B1 | B2 | C1 | C2 | C3 | D1 | D2 | D3 | E1 | E2 | E3 |
| Feichong Camp | 700~800 m | 2~25° | 0.56 | 907 | 1.02 | 2.27 | 1.46 | 0.22 | 0.2 | 0.61 | 0.52 | 0.81 | 0.51 | 0.41 | 0.55 |
| Dangrong Camp | 700~800 m | 2~25° | 0.58 | 963 | 2.31 | 1.53 | 1.37 | 0.13 | 0.13 | 0.47 | 0.21 | 1.75 | 0.37 | 0.33 | 0.53 |
| Zhuzi Camp | 500~600 m | 6~25° | 0.73 | 147 | 1.35 | 1.99 | 1.42 | 0.15 | 0.15 | 0.8 | 0.4 | 1.75 | 0.36 | 0.6 | 0.51 |
| Lizi Camp | 500~600 m | 0~15° | 0.71 | 309 | 2.21 | 1.48 | 1.43 | 0.14 | 0.1 | 0.54 | 0.38 | 0.6 | 0.52 | 0.42 | 0.58 |

Please refer to Figure 1 for the coding legend.

**Author Contributions:** Data curation, Z.N., W.P. and Y.Y.; Formal analysis, W.P.; Funding acquisition, W.C.; Investigation, W.P., Y.Y. and C.H.; Methodology, Z.N. and N.L.; Project administration, N.L.; Resources, W.C. and C.H.; Software, Z.N.; Supervision, N.L.; Writing—original draft, Z.N.; Writing—review and editing, Z.N. and N.L. All authors have read and agreed to the published version of the manuscript.

**Funding:** This research was funded by the Ministry of Education Humanities and Social Sciences Research Planning Fund (21YJAZH042) and the Fundamental Research Funds for the Central Universities from the Central South University.

**Institutional Review Board Statement:** Not applicable.

**Informed Consent Statement:** Informed consent has been obtained from all participants involved in this study.

**Data Availability Statement:** The authors confirm that the data supporting the findings of this study are available within the article.

**Acknowledgments:** Many thanks to Xi Luo from Central South University for his guidance and revision of this article.

**Conflicts of Interest:** The authors declare no conflict of interest.

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
