# Peer review of "Quantitative Research on the Form of Traditional Villages Based on the Space Gene—A Case Study of Shibadong Village in Western Hunan, China"

_sustainability, doi:10.3390/su14148965_

Round 1

Reviewer 1 Report

The paper is extremely well written and structured and informs appropriately in the chosen topic. I honestly have no further comments on its content and I do strongly suggest immediate publication.

Author Response

Dear reviewer:

We sincerely appreciate your careful reading of our paper-"Quantitative Research on the Form of Traditional Villages Based on Space Gene——A case study of Shibadong Village in Western Hunan, China "(ID sustainability-1785943). To your comments, we respond as follows.

Point 1: The paper is extremely well written and structured and informs appropriately in the chosen topic. I honestly have no further comments on its content and I do strongly suggest immediate publication.

Response 1: Thank you very much for your affirmation and encouragement of our article. We will make persistent efforts in the following research and hope to receive your affirmation again. Thank you again from the bottom of our hearts.

Reviewer 2 Report

The Manuscript is about the Space Gene method proposed for the sustainable development of traditional villages. Although the proposed method is remarkable, there are a few points that need to be corrected in the manuscript.

Introduction

The text at the end of the Introduction section should be explained in general without explaining it section by section. Authors should avoid giving chapter numbers.

“The remainder of the article is as follows. The second part is a literature review of traditional village forms. The third part proposes a system model. The fourth part intro- duces the study area and data, the fifth part reveals the results, and the sixth part discusses and summarizes the results.”

2. Literature Review

This part of the study should be enriched by supporting it with related studies in the literature. In this state, it remained weak.

Author Response

Dear reviewer:

We highly appreciate your careful reading of our paper-"Quantitative Research on the Form of Traditional Villages Based on Space Gene——A case study of Shibadong Village in Western Hunan, China "(ID sustainability-1785943). According to the suggestions, we have added and modified the relevant part. All the revisions have been addressed in the Reply and highlighted in the manuscript with a yellow background. A list of changes and responses are as follows.

Point 1: Introduction

The text at the end of the Introduction section should be explained in general without explaining it section by section. Authors should avoid giving chapter numbers.

Response 1: Based on your comments, we have made the following modifications. Please see attached lines 64-67.

The rest of the article is as follows. We first outline the literature on traditional village forms. Next, a quantitative model of traditional village forms is proposed. Then we introduce the study area and validate the model, and finally discuss and summarize the results.

Point 2: Literature Review

This part of the study should be enriched by supporting it with related studies in the literature. In this state, it remained weak.

Response 2: Thank you very much for pointing out the inadequacies of our article. We have made corresponding revisions to the manuscript and added some relevant literature. Please see attached lines 83-69.

Reviewer 3 Report

The Tittle is to long  please look into the suggestion.

please summarize the mathematical formula at the end of the section .

all figures should stated the author and year.

this paper is very rare and good findings 

Author Response

Dear reviewer:

We highly appreciate your careful reading of our paper-"Quantitative Research on the Form of Traditional Villages Based on Space Gene——A case study of Shibadong Village in Western Hunan, China "(ID sustainability-1785943). According to the suggestions, we have added and modified the relevant part. All the revisions have been addressed in the Reply and highlighted in the manuscript with a yellow background. A list of changes and responses are as follows.

Point 1: The Tittle is to long please look into the suggestion.

Response 1: We agree with your point of view, the title of the article is indeed a bit long, and we also wanted to shorten the title, but we found that the article's title reflects the core content of the article, and our titles are all simple words. In addition, we found that other articles in the journal had longer and more cumbersome titles than ours, so we wanted the titles to remain the same.

Point 2: Please summarize the mathematical formula at the end of the section.

Response 2: The mathematical formulas in the article are all used by us, and we have explained the reasons for using this method while introducing each method. For details, please see the yellow background content in the 4.3. Methods section of the article. Please see attached lines 202-206, 213-215, 225-231, 239-240, 245-246 and 254-259.

Point 3: All figures should stated the author and year.

Response 3: We have added source information to all figures and marked revisions in the manuscript. But I looked at many articles in the journal, and none of their figures indicated the year. I think the year may be generated in the system. Please see attachment line 110, line 153, line 286, line 300, line 319, line 329, line 332, line 352, line 376, line 385 and line 400.

Point 4: This paper is very rare and good findings 

Response 4: Thank you very much for your comments and affirmations on our article.

Reviewer 4 Report

Very informative and interesting paper, well written and clear, well structured material. The abstract is well structured representing all main parts of the paper - theoretical considerations, methodology, and research results. The authors have used good and comprehensive literature sources embracing theoretical ideas and empirical experience (knowledge) from different countries and cultures.  The authors have presented well developed theoretical framework and research model showing their analysis logic for their findings.

To improve the paper, I suggest to add some research questions/ hypotheses and to refer to them later in the data analysis part as well as in the conclusion and discussion part.  Also some photos from the village could help to visualize historical, cultural and traditional landscape of the village.

Author Response

Dear reviewer:

We highly appreciate your careful reading of our paper-"Quantitative Research on the Form of Traditional Villages Based on Space Gene——A case study of Shibadong Village in Western Hunan, China "(ID sustainability-1785943). We sincerely thank for raising the important concern and giving us constructive advice. To your comments, we respond as follows.

Point 1: Very informative and interesting paper, well written and clear, well structured material. The abstract is well structured representing all main parts of the paper - theoretical considerations, methodology, and research results. The authors have used good and comprehensive literature sources embracing theoretical ideas and empirical experience (knowledge) from different countries and cultures.  The authors have presented well developed theoretical framework and research model showing their analysis logic for their findings.

Response 1: Thank you very much for your comments and affirmations on our article.

Point 2: To improve the paper, I suggest to add some research questions/ hypotheses and to refer to them later in the data analysis part as well as in the conclusion and discussion part.  Also some photos from the village could help to visualize historical, cultural and traditional landscape of the village.

Response 2: The improvement suggestions you put forward are the main content of our following research. The primary purpose of this article is to test the rationality and practicability of the model, so we spend a lot of space introducing its sources and methods.

We did not present too many research questions for reference in the article. The reason is that our following research will expand the number and categories of our research areas, which will be more conducive to us to put forward some research questions and hypotheses in the article for open discussion. As we have proposed for other research perspectives in the Discussion section of this paper.

Of course, showing some pictures of the village is also our next task.